# Learning Explainable Templated Graphical Models

**Varun Embar** *[*1]          **Sriram Srinivasan** *[1]          **Lise Getoor**[1]

[1]Dept. of Computer Science and Engineering , University of California, Santa Cruz , USA

## Abstract

Templated graphical models (TGMs) encode model structure using rules that capture recurring relationships between multiple random variables. While the rules in TGMs are interpretable, it is not clear how they can be used to generate explanations for the individual predictions of the model. Further, learning these rules from data comes with high computational costs: it typically requires an expensive combinatorial search over the space of rules and repeated optimization over rule weights. In this work, we propose a new structure learning algorithm, *Explainable Structured Model Search (*ESMS*)*, that learns a templated graphical model and an explanation framework for its predictions. ESMS uses a novel search procedure to efficiently search the space of models and discover models that trade-off predictive accuracy and explainability. We introduce the notion of *relational stability* and prove that our proposed explanation framework is stable. Further, our proposed piecewise pseudolikelihood (PPLL) objective does not require re-optimizing the rule weights across models during each iteration of the search. In our empirical evaluation on three realworld datasets, we show that our proposed approach not only discovers models that are explainable, but also significantly outperforms existing state-of-the-art structure learning approaches.

## 1 INTRODUCTION

Templated graphical models (TGMs), a class of probabilistic graphical models that are represented by parameterized potential functions, often use rules or probabilistic constraints to define the model. The templates encode the prob-

---

*[*]Equal contribution

abilistic dependencies between random variables (RVs) and are instantiated many times within the model [Koller and Friedman, 2009]. TGMs have been successfully applied in many domains including computational biology[Segal et al., 2001], knowledge base completion[Jiang et al., 2012], text mining[Beltagy et al., 2014] and computer vision [Aditya et al., 2018]. Learning the components of these models (rules and constraints) directly from the data is known as *structure learning* [Kok and Domingos, 2009, Khot et al., 2011, Mihalkova and Mooney, 2007]. However, it poses several computational challenges. First, the model space is potentially infinite and, even when restricted to be finite, results in a large combinatorial search. Second, approaches that iteratively grow a set of rules require many costly rounds of parameter estimation. Finally, scoring the model often involves computing the model likelihood, which is typically intractable to evaluate exactly.

In addition to predictive performance, there is a growing interest in generating explanations [Wang et al., 2019, Adadi and Berrada, 2018, Zhao et al., 2021, Watson et al., 2021]. Models that provide explanations lead to increased user trust and have also been shown to be more persuasive [Tintarev and Masthoff, 2007, Ribeiro et al., 2016, Alvarez-Melis and Jaakkola, 2018, Doshi-Velez and Kim, 2017, Zhang et al., 2014, Wang et al., 2018]. Explanations can also help isolate and identify incorrect assumptions and biases learned by the model. While TGMs are more interpretable than other large graphical models, generating explanations for individual predictions that satisfy certain desired properties is still challenging. Further, not all rules that are included in the model can be explained to the end user. When learning a model from the data, there may be a need to trade-off accuracy and end-user explainability.

In this work, we propose a novel approach, *explainable structured model search* (ESMS), that learns an explainable templated graphical model automatically from data. Our proposed approach leverages probabilistic soft logic (PSL)[Bach et al., 2017], a TGM defined using a set of weighted first-order logic rules. Unlike other TGMs that use

*Accepted for the 38th Conference on Uncertainty in Artificial Intelligence  (UAI 2022).*

Boolean logic, PSL uses Lukasiewicz logic, a continuous relaxation of Boolean logic, and can incorporate real-valued data such as similarity metrics and confidence scores. Our ESMS approach searches the model space effectively using meta templates that capture common rule patterns. Our proposed structure learning approach utilizes an efficient weight learning strategy that minimizes the need to re-optimize rule weights across models during the search. We also introduce an effective learning objective for PSL that assigns importance weights for rules and eliminates non-informative rules. Our approach uses an explanability score that biases the search to learn explainable models. After learning a model, we generate explanations for each of the predictions using these rules. The use of human-interpretable rules ensures our explanations satisfy the property of *explicitness* [Alvarez-Melis and Jaakkola, 2018]. The continuous nature of inferred values allows us to identify the "true" explanations, a property called *faithfulness*. Further, we extend the property of *stability* for the relational setting and show that the proposed explanation strategy is stable.

The main contributions of our work include: 1) We propose a novel structured search approach that efficiently discovers a templated graphical model using meta templates that best capture the statistical dependencies in the data; 2) We introduce an efficient weight learning strategy based on a piecewise pseudolikelihood objective that allows parallelization and requires weights for a meta template to be learned only once across models; 3) Using an explainabilty parameter, our learning approach generates models that trade-off accuracy and end-user explainability of its predictions; 4) We propose a new Fisher score-based ranking algorithm that identifies the best explanation for a prediction and theoretically show that this is stable; and 5) We empirically show that the discovered models using our proposed approach outperform models generated using state-of-the-art methods.

## 2 RELATED WORK

Our approach builds on a large body of existing work. Here, we give a brief overview of structure learning in templated graphical models and related work in explainability.

**Structure Learning:** Many algorithms have been proposed to learn Markov Logic Networks [Richardson and Domingos, 2006], a class of discrete TGMs. Bottom-up approaches generate informative clauses by using relational paths to capture patterns and motifs in the data [Mihalkova and Mooney, 2007, Kok and Domingos, 2009, 2010]. Most recently, MLN structure learning has been viewed from the perspectives of moralizing learned Bayesian networks [Khosravi et al., 2010] and functional gradient boosting [Khot et al., 2011]. These methods improve scalability while maintaining predictive performance. Structure learning methods specific to a task of interest use inductive logic programming [Mug-

gleton, 1991] to generate clauses which are pruned with L1-regularized learning [Huynh and Mooney, 2008, 2011] or perform iterative local search [Biba et al., 2008] to refine rules with the operations described above. For PSL, a reinforcement learning based approach has been proposed [Zhang and Ramesh, 2019]. Our approach builds on these approaches and the concept of meta templates [Rocktäschel and Riedel, 2017, Wang and Cohen, 2015b, Weber et al., 2019] to learn an model and also generates explanations.

**Explainability:** Explainable AI (XAI) is fast-growing area of research [Ehsan et al., 2021, Arrieta et al., 2020, Gade et al., 2019]. Explainable models can be broadly classified into model-intrinsic methods and model-agnostic methods. Model-intrinsic approaches such as Catherine and Cohen [2016], Kouki et al. [2019], Al-Shedivat et al. [2020] use interpretable models that are easy to explain. Model-agnostic or post-hoc explanations such as Ribeiro et al. [2016], Peake and Wang [2018], Yang et al. [2018] consider the model as a black box and generate explanations from the output. Our proposed approach is a model-intrinsic method that learns an interpretable PSL model. Several gradient-based and perturbation-based explanations have been proposed by Bach et al. [2015], Zeiler and Fergus [2014], Shrikumar et al. [2017], Wolf et al. [2019] for deep learning models. Sundararajan et al. [2017] proposed the notion of integrated gradients that satisfy the properties of sensitivity and implementation invariance. In this work, we propose a similar approach for templated graphical models.

## 3 BACKGROUND

Probabilistic soft logic is a TGM that defines a hinge-loss Markov random field (HL-MRF) [Bach et al., 2017]. The templates are weighted logical rules that encode statistical dependencies and structural constraints. HL-MRFs support modeling of multi-relational data and use a continuous relaxation of discrete logic to generate continuous RVs in the range [0,1]. This allows PSL to incorporate information such as similarity measures. This also makes inference of unobserved RVs efficient and scalable, which is crucial for large scale probabilistic reasoning. PSL has been used successfully in several domains including natural language processing [Beltagy et al., 2014], social media analysis [Johnson and Goldwasser, 2016, Ebrahimi et al., 2016] and recommender systems [Kouki et al., 2015]. As an example, consider the following rule present in a typical recommender system.

$$w : SimItem(I_1, I_2) \wedge Rating(U, I_1) \implies Rating(U, I_2)$$

The rule suggests similar items are rated similarly. Here, $SimItem$ is a **predicate** that encodes the similarity between two items $I_1$ and $I_2$, the predicate $Rating$ encodes the rating assigned to the item by the user $U$ and $w$ denotes the **weight** of the rule which determines its importance. The **variables**

$I_1, I_2, U$ range over the **constants** in a domain. The number of variables in a predicate is called the **arity** of the predicate. The predicate together with the list of variables is called an **atom**. The set of predicates under consideration is denoted by $\mathbf{P}$. Given a set of users {Alice, Bob} and and movies {Legend, Taps}, PSL generates **ground rules** by substituting variables in the rules with constants. An example of a ground rule is as follows:

$$w : SimItem(Legend, Taps) \wedge Rating(Alice, Legend)$$
$$\implies Rating(Alice, Taps)$$

The atoms in a ground rule are called **ground atoms** (e.g. $SimItem$(Legend, Taps)). A **PSL Model** (denoted by $\mathbf{M}$) is a set of weighted rules $\{r_1, r_2, \cdots, r_n\}$. Using the model $\mathbf{M}$ and a set of ground atoms, PSL generates a HL-MRF. PSL associates a RV with each ground atom. RVs with observed values are called **observed RVs** ($\mathbf{X}$) and those with unobserved values are called **unobserved RVs** ($\mathbf{Y}$). These unobserved RVs correspond to the target predicates whose values we wish to infer. For the ground rule mentioned above, let $X_1, Y_1, Y_2$ be the RVs associated with the ground atoms $SimItem(Legend, Taps)$, $Rating(Alice, Legend)$, $Rating(Alice, Taps)$. Then each grounded rule is mapped to a hinge-loss potential $\phi$ using Lukasiewicz logic. For the ground rule mentioned above the hinge-loss potential is given by $\phi(\mathbf{Y}, \mathbf{X}) = max\{X_1 + Y_1 - Y_2 - 1, 0\}^p$. In this work we consider $p = 2$, which results in squared hinge-loss potentials.

Given the set of observed and unobserved RVs $\mathbf{X}, \mathbf{Y}$, and the set of potentials $\mathbf{\Phi}$, PSL defines a probability distribution $\mathbf{X}$ as follows:

$$P(\mathbf{Y}|\mathbf{X}) = \frac{1}{Z(\mathbf{X})} exp(-E(\mathbf{Y}, \mathbf{X}))$$
$$where \; E(\mathbf{Y}, \mathbf{X}) = \sum_j \mathbf{w}_j \mathbf{\Phi}_j(\mathbf{Y}, \mathbf{X}) \quad (1)$$
$$Z(\mathbf{X}) = \int_{\mathbf{Y}} exp(-E(\mathbf{Y}, \mathbf{X}))$$

Here, $j$ iterates over all the ground rules, and $\mathbf{w}$ gives the rule weights. The function $E$ is called the **energy function**.

## 4 EXPLAINABLE TEMPLATED GRAPHICAL MODELS

Explanations are human-understandable artifacts that provide qualitative understanding of the relationship between the data, the model's internal state, and the predictions [Ribeiro et al., 2016, Wolf et al., 2019]. Explanations can either be generated a posteriori, where the model is viewed as a black box, or generated by the model internally along with its predictions. A good explanation must satisfy three properties: *explicitness*, *faithfulness* and *stability* [Alvarez-Melis and Jaakkola, 2018]. Explicitness means that the generated

explanation is interpretable by the user. A faithful explanation implies that the generated explanation is relevant to the prediction. Finally, stability means that the generated explanation does not change drastically for small changes in the input features. The predictions in a TGM depend on the ground rules present in the model. Since these ground rules are human-interpretable, they can be used as explanations.

In a non-relational setting, an explanation is typically a function of the input features. In the relational setting, the generated explanations depend on other observed and unobserved RVs. A stable explanation should not change drastically when the values of other RVs change. We refer to this as *relational stability*. We formally define this by extending the framework in Wolf et al. [2019] to a relational setting.

Let $M$ be a model that predicts the values for the unobserved RVs $\mathbf{Y}$ given the observed RVs $\mathbf{X}$, denoted by $M(\mathbf{X}, \mathbf{Y})$. For example, in PSL, the model infers the values of $\mathbf{Y}$ by identifying the mode of the distribution, e.g., $M(\mathbf{X}, \mathbf{Y}) = \arg\max_{\mathbf{Y}} P(\mathbf{Y}|\mathbf{X})$. Let $\mathbf{G_i}$ denote the set of possible explanations for a RV $\mathbf{Y_i}$.

**Definition 1. Explaining function:** An explaining function, denoted by $f$, produces an importance score of an explanation in $\mathbf{G_i}$ for the inferred value of $\mathbf{Y_i}$.

**Definition 2. Relational Stability**: Let $M$ be a model and $f$ be an explaining function. Let $\mathbf{X}, \mathbf{Y}$ be the set of observed and unobserved RVs and $\mathbf{G_i}$ be the set of possible explanations for the RV $\mathbf{Y_i}$. We say that $f$ is stable with respect to $M$, if for any two $\mathbf{X_1}, \mathbf{X_2}$ that differ in a single RV $X_k$ by at most $\epsilon$, $\exists \delta \in \mathbb{R}$ such that:

$$\forall \mathbf{i} \forall g \in \mathbf{G_i}, |f(\mathbf{X_1}, M(\mathbf{X_1}, \mathbf{Y}), g) - f(\mathbf{X_2}, M(\mathbf{X_2}, \mathbf{Y}), g)|$$
$$\leq \delta$$
$$(2)$$

The above definition states that the explaining function score for every explanation across predictions do not vary a lot when the value of one of the observed RVs is changed by a small value.

Having defined relational stability, we now define the task of learning explainable templated graphical models.

**Definition 3. Learning explainable templated graphical models:** Given a set of predicates $\mathbf{P}$ along with a target predicate $P_T \in \mathbf{P}$ that we need to infer, the task of learning explainable templated graphical model involves two subtasks: 1) The **structure learning** subtask involves discovering a templated model $\mathbf{M}$ that is then used to infer the values of $\mathbf{Y}$ that belong to the predicate $P_T$, and 2) The **explanation** subtask involves generating and ranking the explanations for each of the inferred values of $\mathbf{Y}$ using the explanation function $f$ that satisfies the three properties of explicitness, faithfulness and relational stability.

# 5 LEARNING EXPLAINABLE TEMPLATED GRAPHICAL MODELS

Learning an explainable TGM directly from data poses three main challenges. First, even after restricting the rule length and the size of the model, it involves a combinatorial search and the possible set of models is very large. Second, the search over the space of models involves estimating the weights of the rules many times, which is costly. Finally, not all predicates may be interpretable by the end-user.

To overcome these challenges, we introduce the notion of a *meta template* and propose a novel likelihood function, piecewise-pseudologlikehood (PPLL), to learn the weights of the inferred rules. We also incorporate an *explainabilty bias* that learns a more interpretable model.

## 5.1 META TEMPLATE

Meta templates guide the search by capturing common statistical relational patterns present in the data across a wide range of domains. Further, they restrict the search space by ensuring that the domains and ranges of the predicates are taken into consideration. The concept of a meta template has been proposed for tasks such as predicate learning Muggleton et al. [2015], information and relation extraction [Wang and Cohen, 2015a], question answering[Weber et al., 2019] and in Neural Theorem Provers[Rocktäschel and Riedel, 2017].

**Definition 4. Meta template:** A meta template has slots in place of predicates and encodes the variable bindings between the predicates. Filling the slots with predicates results in a rule.

Consider the following meta template that can be used to combine or fuse information from multiple sources: $\_\_(A, B) \implies P_T(A, B)$. Here, $\_\_$ is a slot that can be filled by a predicate that has the same domain and range as the target predicate. For example, in a hybrid recommender system[Kouki et al., 2015], we can incorporate the outputs of standalone recommender systems such as non-negative matrix factorization ($NMF$) and collaborative filtering($CF$) into our model using this meta template. The rule generated by filling the slot with $NMF$ is given by $NMF(U, I) \implies Rating(U, I)$.

We propose four meta templates that capture a wide variety useful patterns in relational domains. Additional meta templates that generate domain-specific rules can also be incorporated into our approach.

**Path Template:** The path template is the most common meta template and can capture relational patterns such as transitivity. Each slot in the template must be filled with a predicate of arity two. A path template of size two has the following structure: $\_\_(A, B) \wedge \_\_(B, C) \implies P_T(A, C)$

For example, the notion of triadic closure used in social network analysis can be generated from the path template and is given by: $Friends(A, B) \wedge Friends(B, C) \implies Friends(A, C)$. Similarly, path templates of size three and higher can be defined.

**Similarity Template:** The similarity template captures the relationship between multiple target instances. Each slot in the template must be filled with a predicate of arity two and has the following structure: $\_\_(A, B) \wedge \_\_(C, A) \implies P_T(C, B)$ For example, similarity functions used in collaborative filtering can be generated from this template and is given by: $SimilarItem(I_1, I_2) \wedge Rating(U, I_1) \implies Rating(U, I_2)$.

**Local Template:** The local template can integrate information from multiple sources and has the following three structures: $\_\_(A, B) \implies P_T(A, B); \_\_(B) \implies P_T(A, B); \_\_(A) \implies P_T(A, B)$ In addition to our earlier hybrid recommender example, consider the case of fusing multiple classifiers such as $RandomForest$ and $NeuralNetworks$ for the task of entity resolution. We could incorporate them into our model by rules such as: $RandomForest(U_1, U_2) \implies SamePerson(U_1, U_2)$

**Prior Template:** For targets where we have no information, we typically want to encode some prior information. This is captured by the prior template and has the following form: $\mathbf{P}_T(A, B) = \{0, 1\}$ By setting different weights to these rules, we can vary the prior value for targets in the range $[0, 1]$.

## 5.2 PIECEWISE PSEUDOLIKELIHOOD

In addition to the rules, we also need to learn the relative weights of these rules in a PSL model. One approach to weight learning involves optimizing the likelihood function. However, the partition function $Z$ in likelihood involves an integration that makes it intractable to compute. To overcome the intractable likelihood score, pseudo-likelihood [Besag, 1975] is commonly used by weight learning methods. For HL-MRFs, the pseudo-likelihood approximates the likelihood as:

$$P(\mathbf{Y}|\mathbf{X}) = \prod_{Y_i \in \mathbf{Y}} \frac{1}{Z_i(\mathbf{Y_{-i}}, \mathbf{X})} \exp(-E_i(\mathbf{Y}, \mathbf{X}))$$

$$\text{where } E_i(\mathbf{Y}, \mathbf{X}) = \sum_{j:Y_i \in \mathbf{\Phi}_j} \mathbf{w}_j \mathbf{\Phi}_j(\mathbf{Y}, \mathbf{X}) \qquad (3)$$

$$Z_i(\mathbf{Y_{-i}}, \mathbf{X}) = \int_{Y_i} \exp(-E_i(\mathbf{Y}, \mathbf{X}))$$

The notation $j : Y_i \in \mathbf{\Phi}_j$ selects ground rules where $Y_i$ appears. However, due to the coupling of the rules, we also need to re-estimate the weights for the same rule in different models. Further, the objective function is non-convex and is hard to optimize.

To overcome these challenges, we propose to use the efficient-to-optimize objective function called **piecewise pseudolikelihood** (PPLL). PPLL has two key properties that makes weight learning highly scalable : 1) with PPLL, the optimal weight of a rule is independent of other rules in the model; and 2) the PPLL objective is convex and admits an inherently parallelizable gradient-based algorithm for optimization.

PPLL was first proposed for weight learning in conditional random fields (CRF) Sutton and McCallum [2007]. For HL-MRFs, PPLL factorizes the joint conditional distribution along both RVs and rules and is defined as:

$$P(\mathbf{Y}|\mathbf{X}) = \prod_{r \in M} \prod_{Y_i \in \mathbf{Y}} \frac{1}{Z_i^r(\mathbf{Y_{-i}}, \mathbf{X})} \exp(-E_i^r(\mathbf{Y}, \mathbf{X}))$$

$$\text{where } E_i^r(\mathbf{Y}, \mathbf{X}) = \sum_{j: Y_i \in \Phi_j^r} \mathbf{w}_j \Phi_j(\mathbf{Y}, \mathbf{X})$$

$$Z_i^r(\mathbf{Y_{-i}}, \mathbf{X}) = \int_{Y_i} \exp(-E_i^r(\mathbf{Y}, \mathbf{X})) \quad (4)$$

The notation $j : Y_i \in \Phi_j^r$ selects ground rules generated from rule $r$ and has $Y_i$. The key advantage of PPLL over likelihood arises from the factorization of $Z$ into $Z_i^r$, which requires only ground rules corresponding to rule $r$ and variable $Y_i$ for its computation. Following standard convention, we optimize the log of PPLL denoted $l_{ppll}(\mathbf{w})$.

We now show that for the log PPLL objective function, performing weight learning on the entire model containing all rules is equivalent to optimizing the weight for each rule independently.

**Lemma 1.** Optimizing $l_{ppll}(\mathbf{w})$ over the set of weights $\mathbf{w}$ is equivalent to optimizing over each $\mathbf{w}_r$ separately.

*Proof.* By the definition of $l_{ppll}(\mathbf{w})$, we have

$$\arg\max_{\mathbf{w}} l_{ppll}(\mathbf{w})$$

$$= \arg\max_{\mathbf{w}} \sum_{r \in M} \sum_{Y_i \in \mathbf{Y}} -E_i^r(\mathbf{Y}, \mathbf{X}) - logZ_i^r(\mathbf{Y_{-i}}, \mathbf{X})$$

$$= \sum_{r \in M} \arg\max_{\mathbf{w}_r} \sum_{Y_i \in \mathbf{Y}} -E_i^r(\mathbf{Y}, \mathbf{X}) - logZ_i^r(\mathbf{Y_{-i}}, \mathbf{X})$$

$$= \arg\max_{\mathbf{w}_r} \sum_{Y_i \in \mathbf{Y}} -E_i^r(\mathbf{Y}, \mathbf{X}) - logZ_i^r(\mathbf{Y_{-i}}, \mathbf{X}) \forall r \in M$$

We optimize $l_{ppll}(\mathbf{w})$ using a projected gradient descent algorithm. The gradient for a rule weight $\mathbf{w}_r$ turns out to be the difference between observed and expected hinge-loss potential summed over corresponding ground rules $\Phi^r$. We can compute observed penalties once and cache their values. Unlike the gradients for likelihood, each expectation term in the PPLL gradient considers a single rule. Thus, when evaluating gradients for weight updates, we use multi-threading

to compute the expectation terms in parallel. The dual advantages of parallelizing and requiring weight learning only once for a rule makes PPLL highly scalable.

## 5.3 EXPLAINABILITY BIAS

Having introduced key components of our structure search, we next turn to explanability. Some predicates are explainable and other are not. As an example, in a recommender system, rules containing predicates such as $SimUser_{Cosine}$ can be explained using sentences such as "*User $U_1$ who is similar to you liked this item $I$*". Other predicates such as latent factor recommendation approaches may be harder to explain to the end-user. We partition the predicates into explainable and non-explainable predicates. Because explanabilty can be subjective, our approach is flexible, and partitions can be tuned to what seems natural at either the domain level, or even for a particular user. Given a partition, we formally define end-user explainability of a rule as:

**Definition 5** ($\alpha$-explainable). A rule $r$ is $\alpha$-explainable if the proportion of explainable predicates in the body of the rule is greater than $\alpha$.

Therefore, if a rule has no end-user explainable predicates in the body then it is a non-explainable (0-explainable) rule and if every predicate in the body of a rule is end-user explainable then it is a fully explainable (1-explainable) rule.

In applications where providing meaningful explanations to the end user is important, we may prefer models with many $\alpha$-explainable rules. A model with many $\alpha$-explainable rules can result in a greater number of predictions that are explainable. However, this might result in a loss of predictive accuracy. To address this trade-off at the model discovery time, we introduce an explainability bias parameter $\gamma \in [0, 1]$ which is the minimum proportion of rules in a model that are explainable and tune it based on the application's need.

## 5.4 EXPLAINABLE STRUCTURED MODEL SEARCH

Algorithm 1 outlines our proposed ESMS algorithm. For each rule in the model, we first sample a template. We then sample predicates for each slot in the template. We add all $\alpha$-explainable rules to the model, and with probability $1 - \gamma$, we add non-$\alpha$-explainable rules rule to the model. A value of 1 for $\gamma$ and $\alpha$ ensures every rule in the model only contains predicates that are explainable and hence all predictions can be explained. This ensures that the generated explanations satisfy the property of *explicitness*. Once all the rules in the model are sampled, we learn the relative importance of these rules by performing weight learning using PPLL. We then evaluate the performance of the model $V(M)$ on the

## Algorithm 1 Explainable Structured Model Search (ESMS)

**Input:** $T$: Rule templates; $L_M$: Max rules; $N$: max iterations;
   $P$: Set of predicates; $\gamma$: Explainability parameter;
**Output:** $M^*$: Explainable model
   $score_{best} \leftarrow -\infty$
   **for** $i \in 1 \ to \ N$ **do**
      $l_M \leftarrow 0$
      $M \leftarrow \phi$
      **while** $l_M < L_M$ **do**
         $r \leftarrow GenerateRule(T, P, \gamma)$
         $M \leftarrow M \cup r$
         $l_M{+}= 1$
      $\mathbf{w} \leftarrow \text{argmax}_{\mathbf{w}} \, l_{ppll}(\mathbf{w})$
      **if** $V(M) > score_{best}$ **then**
         $M^* \leftarrow M$
         $score_{best} \leftarrow V(M)$
   **return** $\mathbf{M}^*$

## Algorithm 2 Generate Rule(T, P, $\gamma$)

**Input:** $T$: Rule templates; $P$ Set of predicates; $\gamma$: Explainability parameter
**Output:** $r$: a rule
   $RuleFlag \leftarrow False$
   **while** RuleFlag is False **do**
      $t \sim Unif(T)$
      **for** Slot $s$ in $t$ **do**
         Sample $p \in P$ that satisfies domain and range constraints of the variables.
         $r(s) \leftarrow p$
      **if** $r$ is $\alpha$-explainable **then**
         $RuleFlag \leftarrow True$
      **else**
         $g \sim Unif([0, 1])$
         **if** $g \geq \gamma$ **then**
            $RuleFlag \leftarrow True$
   Return $r$

training data. We repeat this process $N$ times and return the best performing model as the final model.

# 6 GENERATING EXPLANATIONS

We now describe our approach to generate explanations for the PSL model's predictions on new data, after we have learned a model using the ESMS approach. The unobserved values are inferred by maximizing the likelihood of the graphical model. The values of the unobserved target RVs $\mathbf{Y}$ depend on all the ground rules they are present in. We can either display these ground rules directly to the user or use a translation system, that takes as input a ground rule and outputs sentences in natural language or pictori-

ally as described in Kouki et al. [2019]. Thus, the set of explanations for a target RV $\mathbf{Y_i}$ (denote by $\mathbf{G_i}$) is given by $\{\phi : \phi \in \mathbf{\Phi} \wedge \mathbf{Y_i} \in \phi\}$.

However, this set is usually large and not all are explanations are equally important. To ensure *faithfulness*, we measure the importance of each ground rule to the inferred value and display the most important rule to the user. We define an explaining function $f$ to score the the importance of ground rules.

**Definition 6.** The explaining function $f : (\mathbf{X}, \mathbf{Y}, \phi) \to \mathbb{R}$ scores the importance of a ground rule $\phi \in \mathbf{G_i}$ with respect to a RV $\mathbf{Y_i}$. It is given by the norm of the first partial derivative of the ground rule at the inferred value $y$, i.e:
$$f(\mathbf{X}, \mathbf{Y}, \phi) = \left\| \frac{w \partial \phi(\mathbf{X}, \mathbf{Y})}{\partial \mathbf{Y_i}} |_y \right\|$$

Unlike other gradient-based approaches such as integrated gradients Sundararajan et al. [2017] where it can be challenging to prove stability, we show in the next subsection that our approach is stable as defined in Section 4.

## 6.1 STABILITY OF THE EXPLANATION FUNCTION

We first observe that the energy function $E$ is a summation of squared hinges and hence $E$ is convex. Further, the prior template described in Section 5 acts a regularizer of $\mathbf{Y}$ and hence $E$ is strongly convex. This was also noted by London et al. [2016].

We state two lemmas that show the change in the optimal energy function is bounded when the value of one of the observed RV ($\mathbf{X}$) is changed (Lemma 2) and this bounds the change is the values of the unobserved RVs $\mathbf{Y}$s (Lemma 3). The proofs for these lemmas are given in the supplementary material.

**Lemma 2.** For a graphical model $G$ with a set of potentials $\mathbf{\Phi}$, let $Q_i$ denote the number of potentials that involve $\mathbf{X_i}$, and let $Q_G \triangleq \max_i Q_i$. Let $\|\mathbf{w}\| < R$. Let $\mathbf{X}, \mathbf{X}' \in \mathcal{X}$ differ at a single coordinate $i$ by at most $\epsilon$. Then, for $\dot{\mathbf{Y}} \triangleq \text{argmin}_{\mathbf{Y}} E(\mathbf{Y}, \mathbf{X})$ and $\dot{\mathbf{Y}}' \triangleq \text{argmin}_{\mathbf{Y}} E(\mathbf{Y}, \mathbf{X}')$, $\left\| E(\dot{\mathbf{Y}}', \mathbf{X}) - E(\dot{\mathbf{Y}}', \mathbf{X}') \right\| \leq \epsilon R \sqrt{Q_G}$

**Lemma 3.** Let $E : (\mathcal{Y}, \mathcal{X}) \to \mathbb{R}$ be $\kappa$-strongly convex, and let $\dot{\mathbf{Y}} \triangleq \text{argmin}_{\mathbf{Y}} E(\mathbf{Y}, \mathbf{X})$ and $\dot{\mathbf{Y}}' \triangleq \text{argmin}_{\mathbf{Y}} E(\mathbf{Y}, \mathbf{X}')$, where $\mathbf{X}, \mathbf{X}' \in \mathcal{X}$ differ at a single RV $\mathbf{X_i}$. Then, $\left\| \dot{\mathbf{Y}}' - \dot{\mathbf{Y}} \right\|^2 \leq \frac{2}{\kappa} \left\| E(\dot{\mathbf{Y}}', \mathbf{X}) - E(\dot{\mathbf{Y}}', \mathbf{X}') \right\|$.

We now state a lemma that shows that the change in the explaining function score for a ground rule $\phi \in \mathbf{G_i}$ denoted by $f(\mathbf{X}, \mathbf{Y}, \phi)$ is bounded.

**Lemma 4.** For an explanation $\phi \in \mathbf{G_i}$, let the explaining function $f$ be defined as $f(\mathbf{X}, \mathbf{Y}, \phi) = \left\| \frac{w \partial \phi(\mathbf{X}, \mathbf{Y})}{\partial \mathbf{Y_i}} |_y \right\|$. Let

$\mathbf{X}, \mathbf{X}' \in \mathcal{X}$ differ at a single RV $\mathbf{X_i}$ by at most $\epsilon$. Let $\|\mathbf{Y} - \mathbf{Y}'\| < B$ for any two $\mathbf{Y}, \mathbf{Y}' \in \mathcal{Y}$ and $\|\mathbf{w}\| < R$. Then $\|f(\mathbf{X}, \mathbf{Y}, \phi) - f(\mathbf{X}', \mathbf{Y}', \phi)\| \leq 2R(\epsilon + B)$

We now prove that the explaining function $f$ is stable.

**Theorem 1.** *The explaining function $f$ is stable with respect to $M(\mathbf{X}, \mathbf{Y})$.*

*Proof.* From Lemma 2 and Lemma 3, for any $\mathbf{X}, \mathbf{X}' \in \mathcal{X}$ that differ in a single RV $\mathbf{X_i}$ by at most $\epsilon$, we have: $\left\| \dot{\mathbf{Y}}' - \dot{\mathbf{Y}} \right\| \leq \sqrt{\frac{2}{\kappa} R \epsilon \sqrt{Q_G}}$

From Lemma 4, we have $\left\| f(\mathbf{X}, \dot{\mathbf{Y}}, \phi) - f(\mathbf{X}', \dot{\mathbf{Y}}', \phi) \right\| \leq 2R(\epsilon + \sqrt{\frac{2}{\kappa} R \epsilon \sqrt{Q_G}})$

## 7 EXPERIMENTAL EVALUATION

We investigate the following research questions empirically: RQ1) What is the predictive accuracy of models discovered by ESMS ? RQ2) What is the impact of the explainability parameter $\gamma$ on end-user explainability? RQ3) How well can the predictions be explained?

**Datasets:** We evaluate the predictive accuracy of the discovered models on an entity resolution dataset and two recommendation datasets. Further, for the recommendation datasets, we evaluate the generated explanations. More details are given in the supplementary material.

**CORA :** This is an entity resolution dataset containing 10 predicates such as the title, venue, author, words in the title and authors that refer to the same entity. The task is to predict publication pairs that refer to same entity.

**YELP :** This is a restaurant recommendation dataset containing 34,454 users, 3,605 restaurants, 8,512 friendship links and 99,049 observed ratings.

**LASTFM :** This is a music artist recommendation dataset containing 1,892 users, 17,632 music artists, 12,717 friendship links and 92,834 observed ratings.

Both the recommendation datasets contain a total of 21 relations such as user and item similarities, and the output of external classifiers such non-negative matrix factorization (NMF). For both datasets, the task is to predict the unobserved ratings. We classified the relations that encode similarity between users or items as explainable and other relations such the output of latent factor models such as NMF as non-explainable; in the end, 15 of the 21 relations were classified as explainable to the end-user. To prevent the generation of a quadratic number of user-item pairs, we perform *blocking*. Blocking restricts the rating pairs by identifying the *important pairs* using a simple heuristic. We use the splits from Kouki et al. [2015].

**Approaches:** We evaluate by comparing the following structure learning methods:

**BOOST[Khot et al., 2011]:** This is a state-of-the-art structure learning approach for MLNs. It uses Friedman's functional gradient boosting algorithm to generate a series of relational regression problems, which in turn are used to generate the rules in the model. We use the code of Khot et al. [2011] with the recursion flag set to True. BOOST uses Boolean logic, so we round the values of the ground atoms to 1 if the value is greater that 0.5, and 0 otherwise. We learn 10 trees and combined the rules across the trees to generate a PSL model. We use the same weights learned by the BOOST approach. Since PSL only allows positive weights, we truncate negative weights to 0. In addition, we also evaluate a model with the weights learned using the PPLL objective (BOOST$_{PPLL}$). Here, we considered all rules discovered by BOOST including rules with negative weights.

**PRA[Gardner and Mitchell, 2015]:** PRA is a relational path finding algorithm that identifies paths that connect unobserved pairs by performing random walks. We use the code of Gardner and Mitchell [2015] to identify paths of length up to three. We then convert these paths to PSL rules. We learn the rule weights using our proposed PPLL weight learning method. We considered all rules including rules with negative weights.

**ESMS**[1]: Our proposed approach that performs a structured search to learn an explainable PSL model. We use the rule templates described in 5. We set the maximum number of rules in a model to 15, maximum iterations to 100 and $\gamma = 0$.

### 7.1 PREDICTIVE PERFORMANCE OF ESMS

We evaluate [RQ1] by comparing the predictive accuracies of BOOST, BOOST$_{PPLL}$, PRA and ESMS . We compute the positive class AUPR for the CORA dataset. For the recommendation datasets we compute the mean squared error (MSE) and mean absolute error (MAE) by rescaling the ratings between $[0, 1]$. Table 1 shows the mean and standard deviation of the metrics computed across the 5 folds. We perform a paired t-test to measure significance and the numbers in bold are statistically significant with $p < 0.05$ . First, we observe that the ESMS approach outperforms both versions of BOOST and PRA on the recommendation datasets. On the entity resolution dataset, it outperforms PRA and is comparable to BOOST. PRA can only discover rules that are paths and this limitation hurts the performance of the model. We next observe that the BOOST models perform better than the BOOST$_{PPLL}$ model. The BOOST method did not learn any collective rules such as: $Rating(A, B) \wedge SimItem(B, C) \implies Rating(A, C)$. These content-based rules are important for the recommender system performance. As a result, the ESMS per-

---

[1]Code, model, and data available at https://github.com/linqs/embar-uai22

|  | CORA | YELP | | LASTFM | |
|---|---|---|---|---|---|
|  | AUPR | MAE | MSE | MAE | MSE |
| BOOST | 0.700 (0.163) | **0.196** (0.008) | 0.079 (0.007) | 0.279 (0.058) | 0.11 (0.044) |
| BOOST$_{PPLL}$ | 0.651 (0.186) | 0.212 (0.012) | 0.092 (0.013) | 0.257 (0.046) | 0.0941 (0.058) |
| PRA | 0.622 (0.169) | 0.2005 (0.0004) | 0.086 (0.0004) | 0.186 (0.001) | 0.048 (0.0004) |
| ESMS | 0.684 (0.148) | **0.193** (0.003) | **0.065** (0.008) | **0.177** (0.070) | **0.043** (0.0004) |

Table 1: **Metrics:** Our ESMS approach significantly outperforms other approaches on recommendation datasets and is comparable to BOOST on CORA. Numbers in bold are statistically significant with $p < 0.05$.

forms better than BOOST. The learned models for all approaches are given in the supplementary material.

ESMS discovered social rules such as $Friends(U_1, U_2) \wedge Rating(U_1, I_1) \implies Rating(U_2, I_1)$, similarity rules such $SimItem_{Pearson}(I_1, I_2) \wedge Rating(U_1, I_1) \implies Rating(U_1, I_2)$. Further, the model incorporates external systems such as Bayesian Probabilistic Matrix Factorization (BPMF) with rules such as $BPMF(U_1, I_1) \implies Rating(U_1, I_1)$.

## 7.2 TRADE-OFF BETWEEN PREDICTIVE ACCURACY AND EXPLAINABILITY

We evaluate [RQ2] by investigating the impact of the explainability parameter $\gamma$ on a model's predictive accuracy and end-user explainability. For each prediction, we generated a ranked list of ground rules in $\mathbf{G_i}$ and compute *mean explainable precision*(MEP@K) [Abdollahi and Nasraoui, 2016] that represents that fraction of ratings that are explainable. MEP@K is defined as $\frac{1}{|\mathbf{Y}|} \sum_{i=1}^{|\mathbf{Y}|} (\mathcal{E}_k(\mathbf{G_i}))$, where $\mathcal{E}_k(\mathbf{G_i})$ is 1 if one of the top-K ranked rules in $\mathbf{G_i}$ is explainable and zero otherwise. We consider a rule to be explainable if it contains at least one explainable predicate ($\alpha = 0.25$). We modified $\gamma$ from 0 to 1 and computed the MEP@1 and MSE of all generated models.

Fig. 1 shows the change in MSE and MEP@1 as we vary $\gamma$ for the LASTFM dataset. We observe that, not surprisingly, we generate models with more explainable rules as we increase $\gamma$. However, the MSE also increases slightly. This is due to the model not containing non-explainable rules such as latent factor models that have high predictive accuracy. We found a similar pattern on the YELP dataset.

### 7.2.1 Analysis of Explanations

We evaluate [RQ3] by analyzing the MEP for all models at $K = \{1, 2, 3\}$. Fig. 2 shows the MEP@K for various approaches. As we increase the value for K, the MEP value increases for all approaches. For ESMS, we get a MEP of 1 for $\gamma > 0.7$ for all $K$. PRA has MEP close to 0.9 due to the large number of rules in the model. BOOST starts with MEP close to 0.5 at $K = 1$ but increases rapidly as we increase $K$.

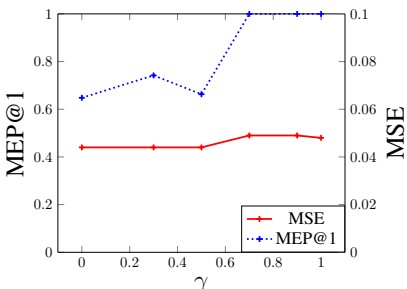

Figure 1: **MEP vs MSE for LASTFM:** As $\gamma$ increases, the models become more explainable and have a slightly higher MSE.

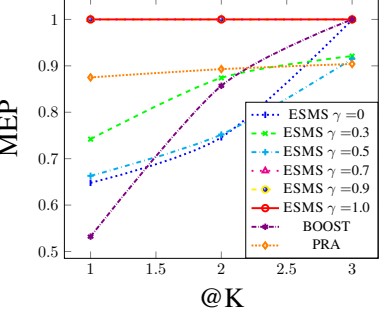

Figure 2: **MEP @ K for LASTFM:** MEP increases for all approaches as we increase $K$. ESMS with $\gamma > 0.7$ outperforms BOOSTand PRA.

As a concrete example of our results, we look at an example of the most important explanation identified by our approach for a rating in the LASTFM dataset. For the pair $(User12, Artist5)$ ESMS identified the most important rule as: $MF(User12, Artist5) \rightarrow Rating(User12, Artist5)$ when $\gamma$ was set to 0. This is a non-explainable rule. When we changed $\gamma = 1$, the most important ground rule became: $Rating(User12, Artist29) \wedge SimItem_{jaccard}(Artist29, Artist5) \rightarrow Rating(User12, Artist5)$. This is explainable.

## 8 CONCLUSION

We proposed an efficient approach to learning explainable templated graphical models that trades off between per-

formance and explainability. Our explanation framework satisfies the properties of explicitness, faithfulness and stability and our search algorithm integrates efficient structure and weight learning. We show that we can learn more explainable model then existing SOTA approaches without compromising much on accuracy. Our work suggests several future directions. Latent predicates are crucial for improving model performance, and we plan to extend our approach to handle them. In addition, we could incorporate end-user preferences into the explanation ranking.

# 9 ACKNOWLEDGEMENTS

This work was partially supported by the National Science Foundation grants (CCF-1740850, CCF-2023495) and an unrestricted gift from Google.

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
