# OpenReview forum: "Learning Explainable Templated Graphical Models"
_auai.org/UAI/2022/Conference — UAI 2022 Poster_

### Official Review · Reviewer_6e3s · 2022-04-09

**Q2(1) Originality/Novelty:** 4
**Q2(2) Significance/Impact:** 4
**Q2(3) Correctness/Technical Quality:** 3
**Q2(6) Clarity Of Writing:** 2
**Q6 Overall Score:** 7
**Q8 Confidence In Your Score:** 5

**Q1 Summary And Contributions:**

The paper proposes a new structure learning
algorithm, Explainable Structured Model Search
(ESMS), that learns a templated graphical model (in particular Probabilistic Soft Logic)
and an explanation framework for its predictions.
ESMS uses a novel search procedure to efficiently
search the space of models and discover models
that trade-off predictive accuracy and explainability.
The paper introduces the notion of relational stability
and proves that the proposed explanation framework
is stable.

**Q2 Assessment Of The Paper:**

More detailed information regarding each of these aspects is given below:

**Q2(4) Quality Of Experiments (Optional):**

3: Good: The experimental evaluation is adequate, and the results convincingly support the main claims.

**Q2(5) Reproducibility:**

4: Excellent: Key resources (e.g., proofs, code, data) are available and key details (e.g., proof sketches, experimental setup) are comprehensively described for competent researchers to confidently and easily reproduce the main results.

**Q3 Main Strengths:**

The paper proposes a structure learning algorithm for PSL that is fast and takes into account the goal of explainability
from the start.

The paper proves that the explanation produces are stable, i.e., they do not change drastically for small changes of the input features.

The experiments compare the approach with meaningful baselines and show evidence of the thesis of the paper.

**Q4 Main Weakness:**

The presentation has a few problems, with some imprecisions and unclear points.

**Q5 Detailed Comments To The Authors:**

The idea of developing a structure learning algorithm that is both fast and geared towards explainability is novel to the best of my knowledge. Moreover, the approach ensures that explanations are stable, a feature not often seen in explainability papers.

The paper is going to have an impact in the broad field of AI concerned with explainability, besides the subfield of Explainable
templated graphical models.

The paper is mostly technically sound except for a few of points:
1) in the proof of theorem 1, in my opinion ||Y'-Y||\leq \sqrt{2/\kappa\epsilon R \sqrt{Q_g}}, so R is inside the \sqrt{} and not outside.
2) in the proof of Lemma 3, in the computation of the derivative ||\deriv\phi(X,Y)/\deriv Y_1|| is really an absolute value, as \phi is a scalar so the derivative is a scalar as well. Moreover, I cannot see why ||\deriv max{C_x X+C_y Y-c,0}/\deriv Y_1 |_y|| is 1
3) You say in definition 7 that function E is \kappa strongly convex but do not sate the value of \kappa and the fact that the prior templates act as regularizer is not clear to me, it should be proved in a lemma.

The reproducibility is excellent, with the code available upon acceptance and all the details of the experimentation reported in the appendix.

The experiments are extensive and consider meaningful baselines. The results on the quality of the solution look good as well as the results on the explainability of the solution in comparison with the baselines.
A few things are not clear.

In table 1 you say that the numbers in bold are statistically significant according to a paired t-test. However, the test compares a pair of results, what is the other result each number in bold is compared with?

Moreover, you state "BOOST
uses Boolean logic, so we round the values of the ground
atoms to 1 if the value is greater that 0.5, and 0 otherwise." do you mean you do so for ESMS, right? Since for BOOT it would not be meaningful.

The sentence "For
the recommendation datasets we compute the mean squared
error (MSE) and mean absolute error (MAE) by rescaling
the ratings between [0, 1]." is not clear: since you use the predicate Rating(User, Item), I thought the rating is given by the value of the atom, so a number in [0,1], so I don't understand the need to rescale the rating.

"We consider a rule to be explainable
if it contains at least one explainable predicate (α = 0.25).": Since there are at most two predicates in the body of the templates, \apha should be 0.5.

The presentation can be improved, with some grammar errors and imprecision, see below for a list.

In Section 5.1 on meta templates you should cite the work on meta interpretive learning by Muggleton et al.
Imprecisions:

Page 4: "a slot that can
be filled by a predicate that has the same domain and range
as the target predicate" do you mean that in P_T(A,B) A is the domain and B is the range? This is not standard terminology,
as a predicate is a function from the cartesian product of the sets of allowed values for their arguments to {0,1}, or, in the case of fuzzy logic, [0,1]. Also in Algorithm 2.

In the discussion on prior templates, P_T(A,B)={0,1} does not look like a rule. maybe better
true<->P_T(A,B) or false <->P_T(A,B)

Page 5: "In applications where providing meaningful explanations to
the end user is important, we may prefer models with high
α-explainable. A model with high α-explainable rules" better
"In applications where providing meaningful explanations to
the end user is important, we may prefer models with many
α-explainable rules. A model with many α-explainable rules"

"A value of 1 for γ ensures every rule in the model
only contains predicates that are explainable and hence all
predictions can be explained." this is true only if \alpha is set to 1

In Algorithm 2, the last if should have the condition g \geq \gamma to ensure that the rule is added anyway with prob 1-\gamma

Appendix F: "mode : recursive_samebib(‘paper,+paper).
mode : recursive_sametitle(title,+title)." the mode is missing from the first argument.

"BPMF(U1, I1)": what is BPMF?

Notation problems:
you are not consistent in using bold for vectors of random variables and non bold for individual random variables: in the left column of page 3 you use non bold capital letters for individual random variables while in the rest of the paper you use bold also for individual variables.

What is \cal X in def 2 and elsewhere? You should either define it or remove \in \cal X as it does not seem needed

Grammar:
Page 2 "confidence scores Our" is missing a period
Page 3: "to the target predicates who values" whose
"they can used as explanations." missing be
Page 4: "( NMF)" remove space
Page 6: "depend on the all the ground rules there are present
in."->"depend on the all the ground rules they are present
in."
Page 7: "True.BOOST" space missing

**Q7 Justification For Your Score:**

Interesting approach for scalable learning of explainable models in PSL, good experimental results. The presentation must be improved and a few things clarified.
-------
The authors' response clarified many of my comments so I raised by score to Accept.

**Q9 Complying With Reviewing Instructions:**

1: Yes.

---

### Official Review · Reviewer_biE9 · 2022-04-12

**Q2(1) Originality/Novelty:** 3
**Q2(2) Significance/Impact:** 3
**Q2(3) Correctness/Technical Quality:** 3
**Q2(6) Clarity Of Writing:** 3
**Q6 Overall Score:** 6
**Q8 Confidence In Your Score:** 3

**Q1 Summary And Contributions:**

In this work, the authors have proposed a novel learning algorithm for template graphical models. These models are explainable and therefore, human interpretable. The authors achieve scalability in this work by restricting the possible number of templates to four and learning predicates around these four templates. The proposed algorithm can produce explanations that can be tuned based on the requisite degree of interpretability.

**Q2 Assessment Of The Paper:**

More detailed information regarding each of these aspects is given below:

**Q2(4) Quality Of Experiments (Optional):**

3: Good: The experimental evaluation is adequate, and the results convincingly support the main claims.

**Q2(5) Reproducibility:**

2: Fair: Key resources (e.g., proofs, code, data) are unavailable but key details (e.g., proof sketches, experimental setup) are sufficiently well-described for an expert to confidently reproduce the main results.

**Q3 Main Strengths:**

This paper presents a novel idea to handle symbolic data.
It is definitely a step forward in the realm of Explainable AI and therefore, a significant contribution to the AI community.
The proposed algorithm is easily implementable, provided the authors augment the current work with the details requested below (in the weakness section).
The paper reads very well and the ideas have been written in very presentable manner.


**Q4 Main Weakness:**

The paper is not sufficiently descriptive about how the predicates are learnt from data.
The time complexity as a function of data arity has not been reported. For real-world problems that would use the method proposed in this work, a sufficiently scalable algorithm would be preferable. Therefore, please specify the time complexity and further, support it using experiments with datasets that vary in arity for the predicates.


**Q5 Detailed Comments To The Authors:**

Firstly, in order to solidify this work, a significant addition to it must be the predicate learning algorithm. The overall template graphical model learning algorithm is well-described but the key information regarding learning the predicates for the four kinds of templates is missing. This detail is significant from the point of view of reproducibility and should be included. In addition, a list of predicates learnt using this method would be quite demonstrative.
Secondly, in order to demonstrate scalability of the approach, please include experiments on datasets with large arities and include the time results as well.
Thirdly, if user studies were to be include with this work to reinforce the explainability of the model, it would further enhance this work.


**Q7 Justification For Your Score:**

This work is novel and advances the field. The ideas, after the concerns are addressed, are largely reproducible and the experimental evaluation sufficiently back the theory.

**Q9 Complying With Reviewing Instructions:**

1: Yes.

---

### Official Review · Reviewer_WXT3 · 2022-04-12

**Q2(1) Originality/Novelty:** 2
**Q2(2) Significance/Impact:** 2
**Q2(3) Correctness/Technical Quality:** 3
**Q2(6) Clarity Of Writing:** 4
**Q6 Overall Score:** 7
**Q8 Confidence In Your Score:** 4

**Q1 Summary And Contributions:**

The paper presents a method for structure learning of Probabilistic Soft Logic (PSL) from data that allows for the selection of explainable rules. The contributions are on a new method for structure learning using meta templates and piecewise pseudo-likelihood, and for a way to tradeoff between explainability and accuracy.

**Q2 Assessment Of The Paper:**

More detailed information regarding each of these aspects is given below:

**Q2(4) Quality Of Experiments (Optional):**

3: Good: The experimental evaluation is adequate, and the results convincingly support the main claims.

**Q2(5) Reproducibility:**

2: Fair: Key resources (e.g., proofs, code, data) are unavailable but key details (e.g., proof sketches, experimental setup) are sufficiently well-described for an expert to confidently reproduce the main results.

**Q3 Main Strengths:**

The paper is very clearly written and organized. The ideas of using meta templates and piece pseudolikelihood are compelling but I am not sure they are entirely novel, particularly the idea of meta templates. In any case, they promise to be impactful since they require a much more tractable computation while still producing good empirical good results. The idea of demanding more explainable predicates in the model is interesting and, according to the empirical evaluation, effective, although relatively simple. The empirical evaluation is compelling and conducted on realistic datasets.

**Q4 Main Weakness:**

The paper does not have significant weaknesses in my opinion, although I would have liked to see more comprehensive evaluation about the trade-off between accuracy and explainability. I was also puzzled by the conversion of alternative structure learning methods to PSL models (see below).


**Q5 Detailed Comments To The Authors:**

I was also puzzled by the described conversion of BOOST and PRA models to PSL rules. Why can't these learned models be evaluated on their own terms? The truncation of BOOST negative weights to 0 seems particularly troublesome.

Figure 1 shows MSE increasing only slightly when explainability increases but does this pattern repeat in other cases as well?

"top-k ranked rules in G_i": is this small k the same as the capitalized K in MPE@K? If so, I would advice to use the same capitalization everywhere.

Typos:

"a ground rules"
"approach, The"
"ground rules there are present in"

**Q7 Justification For Your Score:**

The paper provides at least two interesting and novel contributions (perhaps 3) in two important problems (structure learning and explainability), is clearly written and brings a compelling empirical evaluation.

**Q9 Complying With Reviewing Instructions:**

1: Yes.

---

### Official Review · Reviewer_j7mC · 2022-04-16

**Q2(1) Originality/Novelty:** 3
**Q2(2) Significance/Impact:** 2
**Q2(3) Correctness/Technical Quality:** 3
**Q2(6) Clarity Of Writing:** 3
**Q6 Overall Score:** 6
**Q8 Confidence In Your Score:** 3

**Q1 Summary And Contributions:**

This paper presents a templated graphical model which also enables explanation via sampling explainable predicates. The templated graphical model is a probabilistic soft logic model that can generate a hinge-loss Markov random field. Experiments on an entity resolution dataset and two recommendation datasets are conducted to verify the effectiveness of the proposed model.

**Q2 Assessment Of The Paper:**

More detailed information regarding each of these aspects is given below:

**Q2(4) Quality Of Experiments (Optional):**

3: Good: The experimental evaluation is adequate, and the results convincingly support the main claims.

**Q2(5) Reproducibility:**

2: Fair: Key resources (e.g., proofs, code, data) are unavailable but key details (e.g., proof sketches, experimental setup) are sufficiently well-described for an expert to confidently reproduce the main results.

**Q3 Main Strengths:**

1. Structure learning is an important problem though very hard. The paper is well written. The example from the recommender system helps to understand the terminologies and the overall model.

2. The technical designs are sound. The designed meta templates cover representative ways of composing different models. The hinge-loss MRF is well suited for the task. The approximate inference and learning via piecewise pseudolikelihood are efficient.

3. The controllability of explainability is a great plus to the proposed model. The notion and the proof of relational stability are interesting.


**Q4 Main Weakness:**

1. The search method developed for generating the rule seems to be inefficient, especially when the template space is large. Also, it is unclear how easy it is to sample a predicate that satisfies the domain and range constraints of the variables. It would be great to give more details about how this sampling is done.

2. It would be great to discuss the practical implications of the proposed relational stability. It is hard to get a sense of how useful the proven bound of relational stability in real-world problems is.

3. Could you provide more insights about why your method performs worse on Cora compared to the other baselines but better on the rest of the tasks? Furthermore, bolding multiple entries in Table 1 is very confusing.

**Q5 Detailed Comments To The Authors:**

Although being efficient, there seems to be no guarantee on how good the solution is found by piecewise pseudolikelihood. It would be great to explore the amortized inference techniques for approximate inference.

**Q7 Justification For Your Score:**

Based on my comments above, I tend to accept the paper.


**Q9 Complying With Reviewing Instructions:**

1: Yes.

---

### Decision · Program_Chairs · 2022-05-15

**Decision:**

Accept (Poster)

**Comment:**

Meta Review: The paper introduces a new algorithm for structure learning of probabilistic soft logic (PSL) from data, providing an explanation framework for its predictions. There is a consensus that some of the ideas provided in the paper are novel and will be of interest among the researchers in the field of explainable AI (XAI). The main strenghts are its novelty and originalty, contributing to both structure learning and explainability, and the main weakness are some issues in presenting the results that could addressed by the authors when preparing the camera-ready version.